# Learning Nonparametric Differential Equations via Multivariate Occupation Kernel Functions

## Abstract

Learning a nonparametric system of ordinary differential equations from trajectories in a $d$-dimensional state space requires learning $d$ functions of $d$ variables. Explicit formulations often scale quadratically in $d$ unless additional knowledge about system properties, such as sparsity and symmetries, is available. In this work, we propose a linear approach, the occupation kernel method (OCK), using the implicit formulation provided by vector-valued Reproducing Kernel Hilbert Spaces. The solution for the vector field relies on multivariate occupation kernel functions associated with the trajectories and scales linearly with the dimension of the state space. We validate through experiments on a variety of simulated and real datasets ranging from 2 to 1024 dimensions. The OCK method outperforms all other comparators on 3 of the 9 datasets on full trajectory predictions, and 4 out of the 9 datasets on next point prediction.

## 1 Introduction

### 1.1 Description of the problem

The task of learning dynamical systems derived from ordinary differential equations has garnered a lot of interest in the past couple of decades. In this framework, we are often provided noisy data from trajectories representative of the dynamics we wish to learn, and we provide a candidate model to describe the dynamics. We present a learning algorithm for the vector field guiding the dynamical system that scales linearly with the dimensionality of the dynamical system's state space.

### 1.2 Examples of applications

High-dimensional dynamical systems play a crucial role in various domains. For instance, finite difference methods are commonly used to solve partial differential equations (PDEs), resulting in dynamical systems whose dimensionality scales with the PDE's discretization size Moczo et al. (2004). When the dynamics can be derived from physical principles, modeling them using differential equations leads to favorable generalization.

### 1.3 State of the art

We compare various system identification methods that apply to our relevant experiments, including sparse identification of nonlinear dynamics, reduced order models, deep learning methods, and neural ODEs. These methods learn system dynamics from trajectories in the state space with good accuracy, scale to tens, hundreds, or thousands of state dimensions, and are robust to noise. A detailed description of the comparators is presented in section 4.3.

### 1.4 Main contributions

We propose to use the occupation kernel (OCK) method, that learns a dynamical system and scales linearly with the number of dimensions of the state space. Building upon Rosenfeld et al. (2019a;b); Russo et al.

(2021), our work provides an alternative derivation with the help of vector-valued Reproducing Kernel Hilbert spaces (RKHSs) which allows for the reduction in computational complexity and extensions to physics informed kernels. We also emphasize the simplicity of the algorithm and demonstrate competitive performance on high-dimensional data, as well as noisy data. Finally, in Section 5, we craft an RKHS for learning divergence-free vector fields, demonstrating the versatility of the OCK method.

## 2 Background

### 2.1 RKHS for modeling vector fields

Scalar RKHSs are spaces of real-valued functions that were made popular for their use in constructing the kernelized support vector machine classifier Schölkopf & Smola (2002) chapter III. Scalar RKHSs generalize to vector-valued RKHSs. Vector-valued RKHSs, or simply RKHSs, provide simple non-parametric models - models whose parameter size scales with the training set size - for vector fields. A matrix-valued kernel fully characterizes an RKHS. The choice of this kernel is left to the user and allows for encoding various properties of the functions in the corresponding RKHS. Choosing a kernel with Lipschitz continuous diagonal elements guarantees that all functions in the RKHS $\mathbb{H}$ are Lipschitz continuous. This, in turn, guarantees the local existence and uniqueness of the associated ordinary differential equations $\dot{x} = f(x)$ where $f \in \mathbb{H}$. Furthermore, one may allow the inclusion of physics-inspired constraints into the function space by appropriately choosing the kernel, and an example of a divergence-free RKHS is provided in Section 5. As is well known in the scalar case, RKHSs are spaces of functions amenable to constrained and unconstrained optimization thanks to the representer theorem. Finally, kernels come in two forms: implicit and explicit. The former implicitly characterizes a mapping from $\mathbb{R}^d$ to a Hilbert space, which can be of infinite dimension. Examples include the family of Matérn kernels, which contain the familiar Gaussian and Laplace kernels. The latter explicitly characterizes a mapping from $\mathbb{R}^d$ to $\mathbb{R}^p$ for some $p$. Examples include polynomial kernels as well as random Fourier features. Both kernel types will be used in the experiments in Section 4. We now provide a mathematical presentation of kernels and RKHSs.

### 2.2 Vector valued RKHS

*Definition* 1. Let $\mathcal{M}_{(d,d)}$ be the set of $(d,d)$ matrices, $d \geq 1$. A positive definite matrix-valued kernel, or simply kernel K is a mapping: $\mathbb{R}^d \times \mathbb{R}^d \mapsto \mathcal{M}_{(d,d)}$, such that

1. symmetric: $K(x, x') = K(x', x)^T$, for any $x, x' \in \mathcal{X}$;

2. positive definite: for any $x_1, \ldots, x_n$ in $\mathcal{X}$, and for any $w_1, \ldots, w_n$ in $\mathbb{R}^d$,

$$\sum_{i,j} w_i^T K(x_i, x_j) w_j \geq 0 \tag{1}$$

The simplest kernels are the *separable* kernels

$$K(x, x') = k(x, x')A \tag{2}$$

where $k$ is a positive definite scalar kernel and $A$ is a positive semi-definite matrix. When $A = I$, $\mathbb{H}$ is made of $d$ copies of the RKHS of $k$. All kernels used in our experiments in Section 4 are of this form, while the divergence-free kernel in Section 5 is not a separable kernel. Similarly to the scalar case, there are three ways to characterize a vector-valued RKHS. They are presented in the supplementary material for completeness. Here, we briefly present the construction using the Riesz representation theorem, critical in developing the OCK algorithm.

*Definition* 2. Let $H$ be a Hilbert space of functions $\mathbb{R}^d \to \mathbb{R}^d$ such that for any $x \in \mathbb{R}^d$, the evaluation functional $f \mapsto f(x)$ is continuous, that is, there is a constant $M_x \in \mathbb{R}$, such that

$$||f(x)||_{\mathbb{R}^d} \leq M_x ||f||_H \tag{3}$$

then, using the Riesz representation, for any $v \in \mathbb{R}^d$, there exists a unique $\phi_{x,v} \in \mathbb{H}$ such that

$$v^T f(x) = \langle f, \phi_{x,v} \rangle \tag{4}$$

Define the matrix-valued kernel

$$K_{ij}(x, x') = \langle \phi_{x,e_i}, \phi_{x',e_j} \rangle, i, j = 1 \ldots d \tag{5}$$

where $e_1, \ldots, e_d$ is the natural basis of $\mathbb{R}^d$. Then $K$ is a kernel, $\mathbb{H}$ is the RKHS of $K$ and $\phi_{x,v}(\cdot) = K(\cdot, x)v$.

### 2.3 Occupation kernel functions for vector-valued RKHS

Central to this work is the notion of occupation kernel functions. Let $\mathbb{H}$ be an RKHS of functions from $\mathbb{R}^d$ to $\mathbb{R}^d$ with kernel $K$. Consider a parametric curve $[a, b] \to \mathbb{R}^d$, $t \mapsto x(t)$ and define the functional from $\mathbb{H}$ to $\mathbb{R}^d$, $L_x(f) = \int_a^b f(x(t))dt$. $L_x$ is linear. If $x \mapsto K_{ii}(x, x)$ is continuous for each $i = 1 \ldots d$, then $L_x$ is also continuous[1]. For each $v \in \mathbb{R}^d$, the occupation kernel function $L_x^*(v)$ is then, due to the Riesz representation theorem, the element in $\mathbb{H}$ uniquely defined by the equation $v^T L_x(f) = \langle f, L_x^*(v) \rangle$. Note that for any $w \in \mathbb{R}^d$

$$
\begin{aligned}
w^T L_x^*(v)(y) &= \langle L_x^*(v), K(\cdot, y)w \rangle \\
&= \langle K(\cdot, y)w, L_x^*(v) \rangle \\
&= v^T L_x(K(\cdot, y)w) \\
&= v^T \int_a^b K(x(t), y)w \, dt \\
&= w^T \left[ \int_a^b K(y, x(t))dt \right] v
\end{aligned}
\tag{6}
$$

thus

$$L_x^*(v) = \left[ \int_a^b K(\cdot, x(t))dt \right] v \tag{7}$$

Here, we used the reproducing property equation 42, the symmetry of the inner product, the definition of the occupation kernel and the functional $L_x$, and the symmetry of the kernel. Using the same arguments, we find

$$\langle L_x^*(v), L_y^*(w) \rangle = v^T \left[ \int_a^b \int_a^b K(x(s), y(t))ds\,dt \right] w \tag{8}$$

## 3 Methods

### 3.1 The occupation kernel (OCK) algorithm

We describe the OCK algorithm in two steps. In the first step, we assume that we observe $n$ trajectories or curves in $\mathbb{R}^d$. Each curve is a solution of the ODE defined on a given time interval, with a different known initial condition. We aim to recover the vector field driving this ODE. Assuming that this vector field belongs to an RKHS, and under a penalized least square loss, we provide a solution expressed in terms of occupation kernel functions. In the second step, we relax the hypothesis and assume that snapshots along these trajectories are provided in place of the trajectories. Rearranging these trajectories and replacing the integrals with a numerical quadrature thus provides the proposed numerical solution.

### 3.2 The occupation kernel algorithm from curves

Consider the ODE

$$\dot{x} = f_0(x), x \in \mathbb{R}^d \tag{9}$$

where $f_0 : \mathbb{R}^d \to \mathbb{R}^d$ is a fixed, unknown vector field. Consider also $n$ curves $x_1(t), \ldots, x_n(t)$, from $[a_i, b_i]$ to $\mathbb{R}^d$. Let us design an optimization setting, an inverse problem, for recovering $f_0$ from these trajectories. Let

---

[1]A derivation is provided in the supplementary material

$H$ be an RKHS of continuous vector-valued functions from $\mathbb{R}^d$ to $\mathbb{R}^d$, $\lambda > 0$, a constant, and let us define the functional,

$$J(f) = \frac{1}{n} \sum_{i=1}^{n} \left\| \int_{a_i}^{b_i} f(x_i(t))dt - x_i(b_i) + x_i(a_i) \right\|_{\mathbb{R}^d}^{2} + \lambda \left\| f \right\|_{\mathbb{H}}^{2} \tag{10}$$

By the fundamental theorem of calculus, the first term is minimized when $f = f_0$. However, aside from degenerate cases, this minimizer is not unique. Thus, the second term is a regularization term. The problem of minimizing equation 10 over $\mathbb{H}$ is well-posed. It has a unique solution that can be expressed using the occupation kernel functions as follows:

**Theorem 3.1.** *The unique minimizer of $J$ over the RKHS $\mathbb{H}$ with kernel $K$ is*

$$f^* = \sum_{i=1}^{n} L_{x_i}^{*}(\alpha_i), \alpha_i \in \mathbb{R}^d \tag{11}$$

*where $L_{x_i}^{*}$ is the occupation kernel function of the curve $x_i$ along the interval $[a_i, b_i]$, that is*

$$L_{x_i}^{*}(v) = \left[ \int_{a_i}^{b_i} K(\cdot, x_i(t))dt \right] v \tag{12}$$

*and where the vector $\alpha = (\alpha_1^T, \ldots, \alpha_n^T)^T$ is a solution of*

$$(L^* + \lambda n I)\, \alpha = x(b) - x(a),$$
$$x(a) = \left( x_1(a_1)^T, \ldots, x_n(a_n)^T \right)^T,$$
$$x(b) = \left( x_1(b_1)^T, \ldots, x_n(b_n)^T \right)^T \tag{13}$$

*and $L^*$ is the $(nd, nd)$ matrix made of $(d, d)$ blocks, each defined by*

$$L_{ij}^{*} = \int_{t=a_j}^{b_j} \int_{s=a_i}^{b_i} K\left( x_i(s), x_j(t) \right) ds dt \tag{14}$$

The proof is a straightforward generalization of the representer theorem in RKHSs. It is presented in appendix F. Figure 1 illustrates the result of the theorem.

### 3.3 The occupation kernel algorithm from data

Let us now assume that instead of observing $n$ curves that are solutions of equation 9, we observe $m + 1$ snapshots, sometimes noisy, $z_i = \left( z_i^{(0)}, \ldots, z_i^{(m)} \right)$ at time-points $t_0, \ldots, t_m$ coming from a true trajectory $x_i$, $i = 1 \ldots p$. Firstly, we reshape this data into observations coming from $n = mp$ trajectories, each made of two samples. This is done by viewing every couple of consecutive observations as two observations associated with a single trajectory. In other words, we assume that we observe (possibly with noise) the initial and final points of $n$ trajectories $x_i, i = 1 \ldots n$, that we denote by $z_i = (z_i^{(0)}, z_i^{(1)})$. $z_i$ can therefore be viewed as a matrix of dimensions $d$ by 2.

Secondly, we replace the integral in equation 12 by an integral quadrature, noted $\oint$, and the double integral in equation 14 by a double integral quadrature, noted $\oiint$. The observations $z_i$ are used to compute the quadrature involving trajectory $x_i$. In both cases, we use the trapezoidal rule. The resulting algorithms for learning the vector field and predicting a trajectory given an initial condition are presented in Alg. 1 and Alg 2 respectively. Alg. 1 and Alg 2 are simplified and do not contain the optimizations implemented to reduce the computational complexity (see section 3.4).

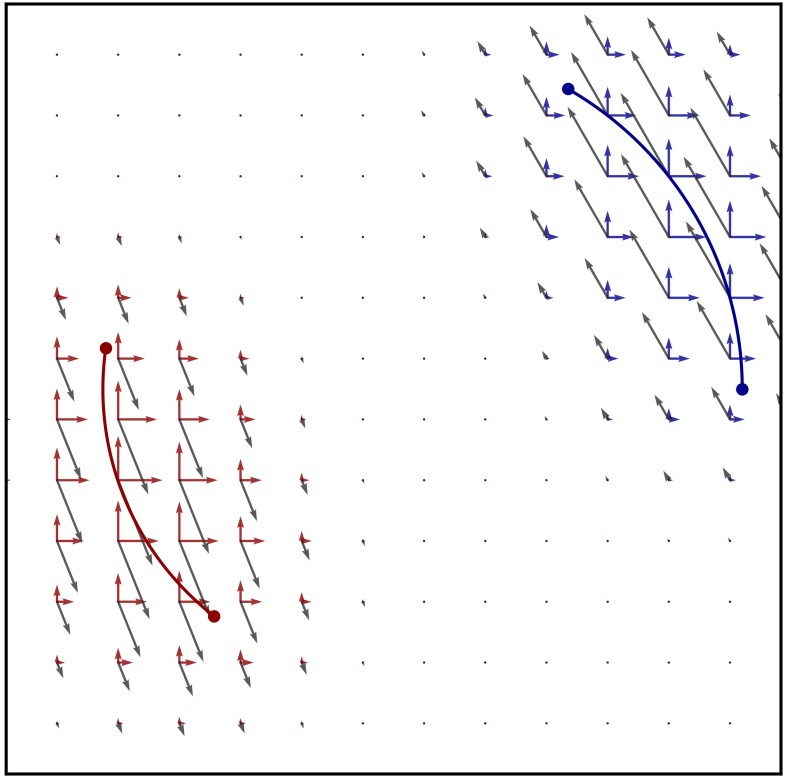

Figure 1: Illustration of the OCK algorithm with the Gaussian separable kernel. We observe the two endpoints of the red and blue trajectories $x_1$ and $x_2$ (running counterclockwise). The horizontal and vertical red vector fields correspond respectively to the occupation kernel functions $L_{x_1}((1,0))$ and $L_{x_1}((0,1))$ and similarly for the horizontal and vertical blue vector fields. The influence of each trajectory is local. The grey vector field is the OCK algorithm solution, a linear combination of the red and blue vector fields.

---

**Algorithm 1** OCK learning: Estimate the parameters defining the vector field.

---

**Require: Training data**. $z_i, i = 1 \ldots n$ ($n$ matrices of dimension $(d, 2)$)

1: Compute $\delta = \left( z_1^{(1)} - z_1^{(0)}, \ldots, z_n^{(1)} - z_n^{(0)} \right)^T \in \mathbb{R}^{n \times d}$

2: **for** $i = 1 \ldots n, j = 1 \ldots n$ **do**

3: $\quad L_{ij}^* \leftarrow \oiint K\left( x_i(s), x_j(t) \right) ds dt$

4: **end for**

5: Solve for $\alpha$ the linear system $\left( L^* + \lambda n I \right) \alpha = \delta$

6: **Return**: $\alpha$

---

---

**Algorithm 2** OCK inference: Generate trajectories given initial conditions.

---

**Require: Training data**. $z_i, i = 1 \ldots n$. ($n$ matrices of dimension $(d, 2)$)
**Require: Output of algorithm 1**: $\alpha = (\alpha_1, \ldots, \alpha_n)^T$.
**Require: Initial conditions**. $p$ vectors :$(y_1^0, \ldots, y_p^0)$
 1: **for** $j = 1 \ldots p$ **do**
 2:     Using a numerical integrator, generate the solution of:

$$\begin{cases} \dot{y}_j = f^*(y_j) \\ y_j(0) = y_j^0 \end{cases}$$

 3:     $f^*(y_j) = \sum_{i=1}^n \left[ \oint K(y_j, x_i(t)) dt \right] \alpha_i$
 4: **end for**

---

### 3.4 Computational complexity

We specialize to the separable kernels with $A = I$, see section 2.2. In such a case, the linear system equation 13 becomes equivalent to solving an $n \times n$ linear system for an $n \times d$ solution. Therefore, this requires $O\left(dn^2\right)$ to construct the kernel matrix, $O\left(n^2\right)$ to estimate the integrals, and $O\left(dn^3\right)$ to solve the linear system. Overall, the OCK algorithm is thus linear in $d$. The space complexity is $O\left(n^2\right)$.

On the other hand, if we let $\psi(x) \in \mathbb{R}^{d \times q}$, then $K(x, y) = \psi(x)^T \psi(y) \in \mathbb{R}^{d \times d}$ is an explicit kernel. In such a case, a $q \times q$ linear system is solved to find $f$, requiring $n$, $d \times q$ by $q \times d$ matrix-matrix multiplications to construct the system. This provides a computational complexity of $O(dnq^2) + O(q^3)$. Further details are provided in appendix D.

## 4 Experiments

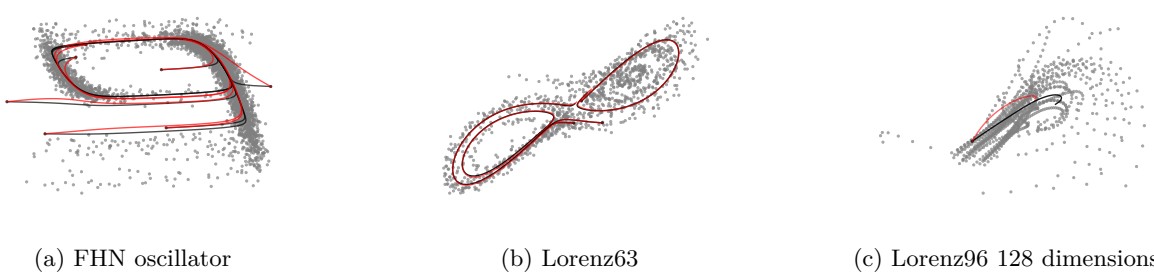

(a) FHN oscillator          (b) Lorenz63          (c) Lorenz96 128 dimensions

Figure 2: The grey points are the training data. The black curves are the test trajectories. The red curves are the predictions on the test set. For data in dimensions higher than two, only the first two dimensions are shown.

### 4.1 Datasets

We test the methods on a diverse set of synthetic and real-world datasets that reflect the challenges of learning systems of ODEs. A summary of the datasets is provided in table 2 in appendix B.

**Noisy Fitz-Hugh Nagumo (NFHN)** The FitzHugh-Nagumo oscillator FitzHugh (1961) is a nonlinear 2D dynamical system that models the basic behavior of excitable cells, such as neurons and cardiac cells[2]. We add considerable noise to this otherwise simple dataset (see figure 7).

**Noisy Lorenz63 (Lorenz63)** The Lorenz63 system Lorenz (1963) is a 3D system provided as a simplified model of atmospheric convection.

---

[2]Further details of the all datasets are provided in appendix B.

**Lorenz96** The Lorenz96 data arises from Lorenz (1995) in which a system of equations is proposed that may be chosen to have any dimension greater than 3. (Lorenz96-16) has 16 dimensions, etc.

**CMU Walking (CMU)** The Carnegie Mellon University (CMU) Walking data is a repository of publicly available data. It can be found at http://mocap.cs.cmu.edu/.

**Plasma** This dataset includes imaging and plasma biomarkers from the WRAP study, see Johnson et al. (2018).

**Imaging** This dataset includes regional imaging biomarkers from the WRAP study, see Johnson et al. (2018).

### 4.2 Kernels

We train the OCK models using the scalar-valued Matérn kernels, including the Gaussian, Laplace, and $C^{10}$ kernels Fuselier et al. (2016). We also train an explicit kernel with 200 Fourier Random features Rahimi & Recht (2007).

### 4.3 Comparable methods

We select competitive methods covering various categories.

**Sparse Identification of Nonlinear Dynamics (SINDy)** SINDy is a well-developed class of data-driven methods for identifying dynamical systems models from trajectories. Brunton et al. (2016) These methods rely on sparse regression techniques to isolate the most relevant terms in the governing equations from a set of candidate functions. SINDy demonstrates robustness for sparse and limited data Kaheman et al. (2020); Fasel et al. (2022).

**Reduced Order Models (ROMs)** ROMs are a class of methods for simplifying high-dimensional dynamical systems by projecting them onto a lower-dimensional space. Dynamic mode decomposition (DMD) is a data-driven method for extracting spatiotemporal patterns and coherent structures from high-dimensional data generated by dynamical systems Tu (2013). eDMD extends DMD to handle nonlinear dynamics by working in a higher-dimensional feature space. Williams et al. (2015). We benchmark with eDMD-RFF Lew et al. (2023), eDMD-Poly Williams et al. (2015), and eDMD-Deep Yeung et al. (2019)

**Deep Learning Methods** Deep learning methods can be used in conjunction with ROMs to learn transformations to lower-dimensional spaces. These methods use deep learning to construct an efficient representation of the dynamical system and to capture nonlinear dynamics Lusch et al. (2018); Yeung et al. (2019); Li et al. (2019). Deep learning methods can also be incorporated into the SINDy framework to identify sparse, interpretable, and predictive models from data Champion et al. (2019); Bakarji et al. (2022). We benchmark with ResNet He et al. (2016) Lu et al. (2021)

**Latent ODEs for Irregularly-Sampled Time Series** The Latent ODE method (also a deep learning method Rubanova et al. (2019)) is an update of the Neural ODEs model introduced in Chen et al. (2018). The core idea is to represent the hidden dynamics of time series data in a continuous latent space using ODEs. Given the latent trajectory, the observations of the time series are assumed to be independent. The latent trajectory dymanics are governed by a neural ODE model. The model uses an encoder-decoder framework where the encoder maps observed data to a latent initial value via an RNN architecture where the hidden states are carried through neural ODEs between times of observations, and the decoder generates the latent trajectory forward in time and predicts future observations.[3]

The hyperparameters for each method are shown in Table 3 in appendix C. All experiments are run in Google Colab using the default settings. The GPU was only enabled (A100, High-RAM) for the deep learning techniques. Lastly, to get an idea of the difficulty of each problem, we compare all methods against the null model, which predicts no change, that is $\mathbf{x}(t) = \mathbf{x}_0$.

---

[3]Latent ODEs was implemented using the Github repository: https://github.com/YuliaRubanova/latent_ode. The subsampling was disabled for the Plasma and Imaging datasets because the training trajectories were too short in these datasets.

### 4.4 Evaluation Method

For each dataset, the trajectories were split as follows: training, validation, and testing. For validation of the OCK method, we used gaussian process hyper parameter tuning (on a log scale) using https://skopt.readthedocs.io/en/stable/. At test time, we integrated the predicted trajectories starting at the given initial conditions and compared to the true trajectories. To measure the performance of each method, we define two types of errors for each trajectory of the test set. We define:

$$\text{Err} = \sqrt{\sum_{i=2}^{n} (t_i - t_{i-1}) \|y_i - \hat{y}_i\|^2} \tag{15}$$

where $y_i$ and $\hat{y}_i$ are the observed and predicted trajectory samples at time $t_i$, respectively. Additionally, we define 1-Err by setting $n = 2$ in equation 15. All the methods were trained, validated, and tested on the same data. Table 2 provides the average Err and 1-Err errors across the trajectories of the test set. To test whether the OCK methods are (statistically) significantly better than comparable methods and vice versa, we first generated a Wilcoxon test p-value for every OCK method (four different kernels) against the other comparable methods. The pairs of observations used in the test are the errors of the two compared methods for every trajectory in the test set. Finally, to generate a p-value comparing all OCK methods together against another comparable method, we used Fisher's method Mosteller & Fisher (1948) to combine the four p-values of the OCK methods. We report a $*$ in Table 1 if the group of OCK methods is significantly better ($p < 0.01$) and † if the compared method outperforms OCK with statistical significance ($p < 0.01$). Otherwise, no method is significantly better than the other.

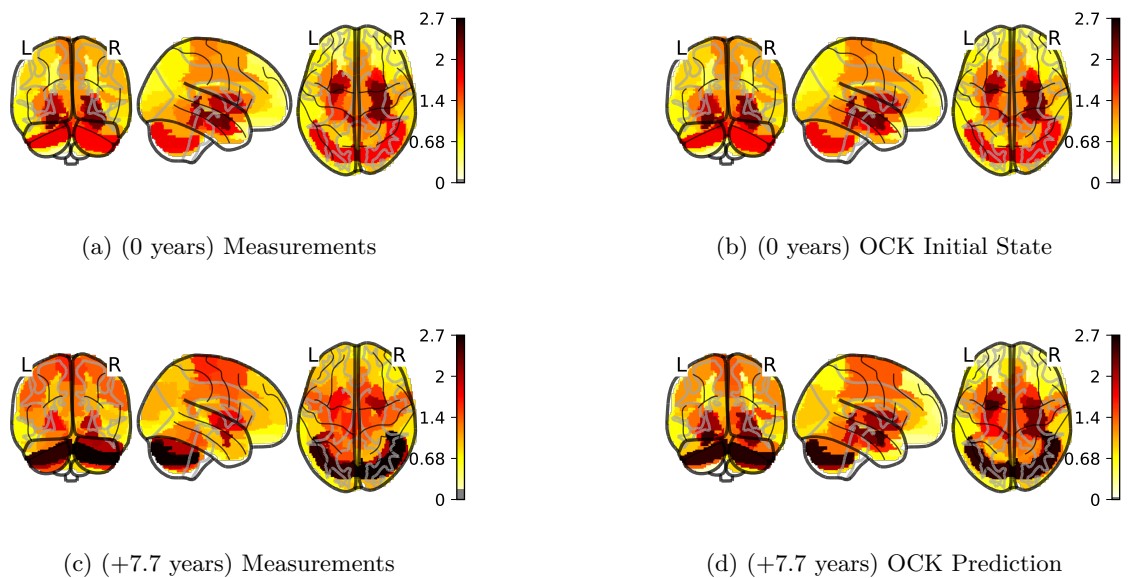

(a) (0 years) Measurements         (b) (0 years) OCK Initial State

(c) (+7.7 years) Measurements         (d) (+7.7 years) OCK Prediction

Figure 3: OCK Predicted Biomarker Values from the imaging dataset.

### 4.5 Evaluation Performance

Examples of the output of the OCK algorithm are presented visually in figures 2 and 3. Table 1 summarizes the prediction errors for each experiment. While no single method outperforms other methods over all datasets, OCK outperforms in 3 of the 9 datasets (on the task of full trajectory prediction, denoted Err), and outperforms all other methods on 4 of the 9 datasets on the task of predicting the next sample (denoted 1-Err). It also performs competitively on the remaining datasets on both tasks. Notably, eDMD-Deep yielded the best results for the CMU experiments. Deep learning models and DMD methods learn the dynamics in a feature space and therefore have less stringent constraints on the dynamics. We believe this is why

Table 1: Dynamical System Estimation Results

**Lorenz63**

|  | Err | 1-Err |
|---|---|---|
| ResNet | 2.07* | .014* |
| eDMD-Deep | 2.91* | .009* |
| eDMD-Poly | 2.22* | .012* |
| eDMD-RFF | 1.93* | .011* |
| null | 2.56* | .011* |
| SINDy-Poly | .99* | **.002** |
| ockG | .66 | .002 |
| ockL | **.65** | .002 |
| ockM | 2.52 | .012 |
| ockF | 2.52 | .012 |
| Lode | 1.49* | .092* |

**NFHN**

|  | Err | 1-Err |
|---|---|---|
| ResNet | 5.94* | .064* |
| eDMD-Deep | 4.54* | .036* |
| eDMD-Poly | 4.10* | .037* |
| eDMD-RFF | 4.01* | .037* |
| null | 9.69* | .043* |
| SINDy-Poly | 1.68* | **.022†** |
| ockG | 1.40 | .029 |
| ockL | 1.11 | .030 |
| ockM | 1.41 | .029 |
| ockF | 1.38 | .028 |
| Lode | **.64†** | .131* |

**Lorenz96-16**

|  | Err | 1-Err |
|---|---|---|
| ResNet | 6.42* | .010* |
| eDMD-Deep | 5.93* | .040* |
| eDMD-Poly | 6.91* | .036* |
| eDMD-RFF | 6.22* | .029* |
| null | 7.50* | .039* |
| SINDy-Poly | .40* | .0003* |
| ockG | .69 | .0007 |
| ockL | **.22** | **.0001** |
| ockM | .22 | .0001 |
| ockF | .22 | .0001 |
| Lode | 3.82* | .138* |

**Lorenz96-32**

|  | Err | 1-Err |
|---|---|---|
| ResNet | 11.56* | .020* |
| eDMD-Deep | 8.79* | .057* |
| eDMD-Poly | 9.67* | .051* |
| eDMD-RFF | 8.10* | .028* |
| null | 10.49* | .056* |
| SINDy-Poly | 8.56* | .037* |
| ockG | .47 | **.0001** |
| ockL | .48 | .0001 |
| ockM | .49 | .0001 |
| ockF | **.47** | .0001 |
| Lode | 6.06* | .217* |

**Lorenz96-128**

|  | Err | 1-Err |
|---|---|---|
| ResNet | 24* | 2.64* |
| eDMD-Deep | 17* | .089* |
| eDMD-Poly | 19* | .109* |
| eDMD-RFF | 16† | .201* |
| null | 21* | .112* |
| SINDy-Poly | 19* | .066* |
| ockG | 17 | .064 |
| ockL | 16 | **.035** |
| ockM | 16 | .040 |
| ockF | 16 | .036 |
| Lode | **15†** | .342* |

**Lorenz96 -1024**

|  | Err | 1-Err |
|---|---|---|
| ResNet | 66.1* | 6.48* |
| eDMD-Deep | 26.9* | .41* |
| eDMD-Poly | 61.5* | .42* |
| eDMD-RFF | 54.2* | .38* |
| null | 67.0* | .16* |
| SINDy-Poly** | NA | NA |
| ockG | 18.0 | **.013** |
| ockL | 17.8 | .014 |
| ockM | 17.9 | .013 |
| ockF | 18.0 | .015 |
| Lode | **15.8†** | 1.04* |

**Plasma**

|  | Err | 1-Err |
|---|---|---|
| ResNet | 4.93* | 2.89* |
| eDMD-Deep | 4.09 | 2.42 |
| eDMD-Poly | 4.01 | **2.39** |
| eDMD-RFF | 4.07 | 2.43 |
| null | 4.00* | 2.41 |
| SINDy-Poly | **3.95** | 2.40 |
| ockG | 3.96 | 2.41 |
| ockL | 3.96 | 2.41 |
| ockM | 3.96 | 2.41 |
| ockF | 3.96 | 2.41 |
| Lode | 4.28* | 2.68* |

**Imaging**

|  | Err | 1-Err |
|---|---|---|
| ResNet | 27.4* | 19.4* |
| eDMD-Deep | 15.6* | 12.5 |
| eDMD-Poly | 15.7 | 12.5 |
| eDMD-RFF | 27.9* | 19.3* |
| null | 16.2* | 12.6* |
| SINDy-Poly | 96.1* | 53.7* |
| ockG | 15.6 | 12.3 |
| ockL | 15.6 | 12.3 |
| ockM | 15.7 | 12.3 |
| ockF | 15.8 | 12.4 |
| Lode | **14.8†** | **11.7** |

**CMU**

|  | Err | 1-Err |
|---|---|---|
| ResNet | 22.6* | .86† |
| eDMD-Deep | 15.7† | 2.03* |
| eDMD-Poly | **14.9†** | .78† |
| eDMD-RFF | 14.9† | **.76†** |
| null | 21.6* | 1.01* |
| SINDy-Poly | 33.8* | .91* |
| ockG | 21 | .89 |
| ockL | 19.6 | .93 |
| ockM | 21.5 | 1.01 |
| ockF | 21.6 | 1.01 |
| Lode | 14.9† | 1.88* |

**Description:** Minimum values are in bold. A $*$ indicates a significant (Fisher's method) p-value ($< 0.01$) in favor of the ock methods in the comparison. A † indicates a significant p-value in favor of the method compared with all ock methods. There is no statistically significant difference otherwise. Our models are labelled as ockG - occupation kernel method with Gaussian kernel, ockM - occupation kernel method with Matern kernel, ockF - occupation kernel method with Random Fourier Features (RFF), and ockL - occupation kernel with Laplace kernel (See section 4.2). We compare against SINDy-Poly, eDMD-Deep, eDMD-Poly, eDMD-RFF, ResNet and Latent Ode (Lode) (See section 4.3). ** No result could be obtained for SINDy-Poly for this dataset.

they outperform with datasets such as the CMU dataset as our model assumes the dynamics arises from a homogeneous system which is just not the case.

The differences observed in table 1 are not always significant. Recall that we use the sign † when a method is better than all the OCK methods. We count 5 models out of 9 with at least one † for Err and only 2 out of 9 for the 1-Err. Note that the OCK algorithm is optimized for the 1-Err since the trajectories of the training set are systematically reshaped in trajectories of two observations, thus it is not surprising that it performs better for this metric.

When comparing the kernels used for the OCK method, the Laplace kernel performs the best overall, consistent with other analyses of this kernel (Geifman et al. (2020)). Note also that the random Fourier features kernel, which approximates the Gaussian kernel, performs well, allowing for linear implementations in $n$, which is the number of examples in the training set.

The trajectories predicted by ockL for NFHN (in blue in figure 6) are visually close to the ground truth (in black) in all but the last case. SINDy-poly (in yellow) is competitive. ResNet provides very irregular trajectories. Latent ODE provides good long-term predictions. However, the initial points do not coincide with the initial points of the ground truth provided for all the algorithms. In other words, while Latent-ODE performs very well on the task of predicting full trajectories, it performs very poorly on the next sample prediction task which is an inherent limitation of some deep learning models like Latent-ODE and Resnet.

We tested OCK with a dataset of 1024 dimensions to show how well it scales. The implementation of SINDy-Poly that we used did not provide a result for this data; see NA in the table 1.

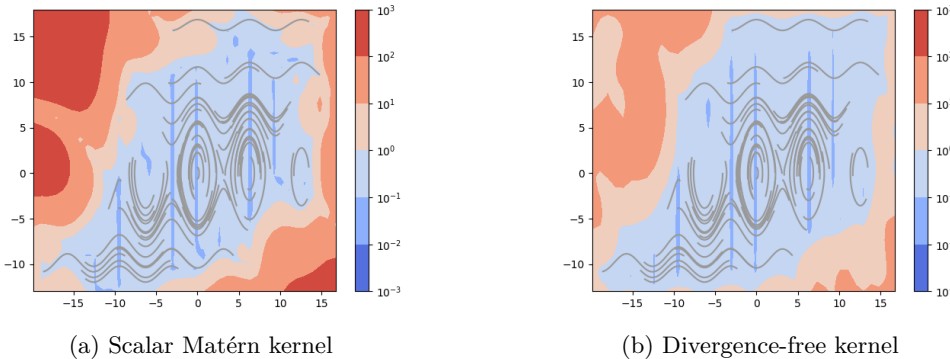

(a) Scalar Matérn kernel          (b) Divergence-free kernel

Figure 4: The magnitude of the error in vector field approximation of the pendulum problem obtained using different choices of kernel along with the training set (gray).

## 5 Learning vector fields with constraints

Divergence-free ($\nabla \cdot v = 0$) and curl-free ($\nabla \times v = 0$) vector fields appear in a diverse set of applications including fluid dynamics Wendland (2009); Fuselier et al. (2016), magnetohydrodynamics McNally (2011), image processing Polthier & Preuß (2003), and surface reconstruction Drake et al. (2022). Accordingly, a significant amount of research has been devoted to the development of curl-free and divergence-free kernel methods in recent years Narcowich & Ward (1994); Lowitzsch (2005); Narcowich et al. (2007); Fuselier (2008); Fuselier & Wright (2009); Gao et al. (2022). We will demonstrate how the occupation kernel method can be adapted to ensure that the recovered vector field analytically satisfies the divergence-free/curl-free constraint for an appropriate choice of matrix kernel $K$. Then, for ease of presentation, we will apply just the divergence-free kernels to both real and synthetic datasets.

### 5.1 Divergence-free kernels

Let $x \in \mathbb{R}^d$ and $d = 2, 3$, and let $\phi(\|x\|)$ be a radial basis function. Then the standard construction Fuselier (2008) for divergence-free and curl-free matrix-valued radial basis functions are

$$\{-\Delta I + \nabla \nabla^T\}\phi(\|x\|) \quad \text{and} \quad -\nabla \nabla^T \phi(\|x\|), \tag{16}$$

respectively, where $\Delta = \sum_{i=1}^{d} \partial^2/\partial x_i^2$ and $[\nabla \nabla^T]_{ij} = \partial^2/\partial x_i \partial x_j$, $1 \le i, j \le d$. For example, if $d = 2$, then

$$\{-\Delta I + \nabla \nabla^T\}\phi(\|x\|) = \begin{bmatrix} -\frac{\partial^2 \phi}{\partial x_2^2} & \frac{\partial^2 \phi}{\partial x_1 \partial x_2} \\ \frac{\partial^2 \phi}{\partial x_2 \partial x_1} & -\frac{\partial^2 \phi}{\partial x_1^2} \end{bmatrix} (\|x\|). \tag{17}$$

Thus, the columns of the matrix-valued radial basis function $\{-\Delta I + \nabla \nabla^T\}\phi(\|x\|)$ are divergence-free. Similarly, we have that the columns of $-\nabla \nabla^T \phi(\|x\|)$ are curl-free.

We select the divergence-free kernel $K(x, y) = \{-\Delta I + \nabla \nabla^T\}\phi(\|x - y\|)$ where our choice of scalar radial basis function is the $C^{10}$ Matérn kernel Fuselier et al. (2016).

### 5.2 Hamiltonian systems

A system

$$x_1' = f(x_1, x_2), \quad x_2' = g(x_1, x_2) \tag{18}$$

is called a *Hamiltonian system* if there exists a function $H(x_1, x_2)$ (called the *Hamiltonian*) for which $f = H_{x_2}$ and $g = -H_{x_1}$. This implies that the vector field $(f, g)$ is divergence-free. Furthermore, along every orbit we have $H(x_1, x_2) = $ constant and any conservative dynamical equation $u'' = f(u)$ leads to a Hamiltonian system where the Hamiltonian coincides with the total energy. For example, every orbit of the conservative system corresponding to the equation describing the motion of a pendulum

$$x_1' = x_2, \quad x_2' = -\frac{g}{\ell} \sin(x_1) \tag{19}$$

satisfies the conservation law

$$H(x_1, x_2) = \frac{1}{2}x_2^2 - \frac{g}{\ell} \cos(x_1) = E \tag{20}$$

for some constant $E$.

### 5.3 Experiments

We test the divergence-free OCK method on the following real and synthetic datasets.

**Pendulum Problem** We generate data using (19) with $\ell = 1$ and $g = 9.8$. This 2-dimensional synthetic dataset consists of 100 trajectories of 50 samples each. This dataset allows us to test whether the OCK method can, simultaneously, learn a known Hamiltonian system while analytically enforcing the divergence free constraint.

**Buoy Data** We obtained a buoy dataset from https://oceanofthings.darpa.mil/data#tab-all, which contains two-dimensional trajectories of buoys submerged in the ocean. We sampled every tenth observation of the raw data and converted the time measurements into years. Trajectories with only one sample were dropped.

### 5.4 Evaluation

The loss was computed using (15) for both datasets. For the pendulum problem, the scalar Matérn kernel provided a loss of $1.5 \times 10^{-1}$ while the divergence-free kernel provided a loss of $1.9 \times 10^{-2}$, an improvement by an order of magnitude. Not only does the divergence-free kernel allow for a physically constrained solution, but it is also more accurate. For the buoy data, the scalar Matérn kernel provided a loss of $3.5 \times 10^1$ while the divergence-free kernel provided a loss of $3.1 \times 10^1$. Thus, despite the presence of noise (which is not necessarily divergence-free), we still achieve a reduction in the loss of about 10%.

Figure 4 shows the error in the computed vector fields of the pendulum problem. A plot of the results for the buoy data problem using the divergence-free kernel, along with the training data, is shown in figure 5.

## 6    Summary and discussion

The implicit matrix-valued kernel formulation has provided us the means to demonstrate that the OCK algorithm, originally proposed by J. Rosenfeld and co-authors, allows for learning the vector field of an ODE, and scales linearly with the dimension of the state space. We have compared the algorithm to a representative selection of competitive algorithms. We have provided convincing experimental evidence of superior performance of the method on many of the datasets we tested, and consistently competitive performance on all datasets. The surprising simplicity of the technique lends to many simple and exciting follow-up ideas for optimizations, generalizations, and explorations. Among other topics, we expect follow-up work to push the state-of-the-art further into the regime of high-dimensional dynamics learning, applications to PDEs, and a thorough analysis of the theoretical convergence of the method.

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

## A   Experiment Code

We created a code capsule on Code Ocean that hosts all the methods used in our experiments for system identification of ordinary differential equations. The code repository allows for the repeatability of our results for a single experiment—Lorenz63—and facilitates the evaluation of the performance of different models on synthetic and real-world datasets. The real world plasma and imaging experiment are restricted data and will not be available for the evaluation.

We provide an *anonymized* zipfile (6.6 MB) export of the code capsule for review at `https://drive.google.com/file/d/1p36HH00dHGLuBeHEfhJyMlfxjv1At5La/view?usp=sharing`. The code can be run locally in a Docker container with instructions in the README.md and REPRODUCING.md. After review, we will publish a de-anonymized capsule available on the Code Ocean platform, as well as a public GitHub repository of the code with a detailed README.md covering the Papers with Code release checklist [4].

To ensure the reproducibility of the experiment, we have tuned the hyperparameters for each method, except for ResNet, which is known to require substantial tuning time. The code repository provides detailed instructions for running each method experiment, including the necessary input data and the corresponding hyperparameter settings. Please note that due to limitations on Code Ocean, we have set the TensorFlow library to use the CPU rather than the GPU, as using the GPU resulted in numerous errors. This may increase the runtime of the experiments. For the paper, we ran some experiments in google colab on A100 GPUs where the training time was much lower.

Upon running the code repository, the experiments will take approximately 1.5 hours to complete. The results of each experiment are provided in the form of trajectory CSV files, capturing the predicted trajectories of the identified systems. These trajectory files can be further analyzed or visualized as desired. Additionally, a summary of the performance of each method is available in the code repository.

## B   Datasets

In table 2, d refers to the number of dimensions of the system, n the total number of trajectories, and m the average number of samples per trajectory.

---

[4]https://github.com/paperswithcode/releasing-research-code

Table 2: Datasets

| Name | Type | d | n | m |
|---|---|---|---|---|
| NFHN | Synthetic | 2 | 150 | 201 |
| Lorenz63 | Synthetic | 3 | 150 | 201 |
| Lorenz96-16 | Synthetic | 16 | 100 | 243 |
| Lorenz96-32 | Synthetic | 32 | 100 | 243 |
| Lorenz96-128 | Synthetic | 128 | 100 | 243 |
| Lorenz96-1024 | Synthetic | 1024 | 100 | 100 |
| CMU | Real | 50 | 75 | 106.7 |
| Plasma | Real | 6 | 425 | 2.95 |
| Imaging | Real | 117 | 231 | 2.26 |

## B.1 Synthetic data

**Noisy Fitz-Hugh Nagumo**   The FitzHugh-Nagumo oscillator FitzHugh (1961) is a nonlinear 2D dynamical system that models the basic behavior of excitable cells, such as neurons and cardiac cells (See figure 7). The system is popular in the analysis of dynamical systems for its rich behavior, including relaxation oscillations and bifurcations. The FHN oscillator consists of two coupled ODEs that describe the membrane potential $v$ and recovery variable $w$,

$$
\dot{v} = v - \frac{v^3}{3} - w + RI
$$
$$
\tau \dot{w} = v + a - bw
$$

(21)

Here, $I$ is the external current input, $a$ and $b$ are positive constants that affect the shape and duration of the action potential, and $\tau$ is the time constant that determines the speed of the recovery variable's response. We added random Gaussian noise of standard deviation 0.12 to this dataset.

**Noisy Lorenz63**   The 3D Lorenz 63 system Lorenz (1963) is a simplified model of atmospheric convection, but has since become a canonical example of chaos in dynamical systems. The Lorenz 63 system exhibits sensitive dependence on initial conditions and parameters, which gives rise to its characteristic butterfly-shaped chaotic attractor. The equations are

$$
\frac{\mathrm{d}x}{\mathrm{d}t} = \sigma(y - x)
$$
$$
\frac{\mathrm{d}y}{\mathrm{d}t} = x(\rho - z) - y.
$$
$$
\frac{\mathrm{d}z}{\mathrm{d}t} = xy - \beta z
$$

(22)

Here, $x$, $y$, and $z$ are state variables representing the fluid's convective intensity, and $\sigma$, $\rho$, and $\beta$ are positive parameters representing the Prandtl number, the Rayleigh number, and a geometric factor, respectively. We added random Gaussian noise with standard deviation 0.5 to this dataset.

**Lorenz96**   The Lorenz96 data arises from Lorenz (1995) in which a system of equations is proposed that may be chosen to have any dimension greater than 3. The chaotic system is defined by:

$$
\frac{dx_k}{dt} = -x_{k-1}x_{k-2} + x_{k+1}x_{k-1} - x_k + F
$$

(23)

where we take $F = 8$ and consider 16, 32, 128 and 1024 dimensional systems. Indices are assumed to wrap so that for an $n$ dimensional system $x_{-2} = x_{n-2}$.

## B.2 Real Data

**CMU Walking**   The Carnegie Mellon University (CMU) Walking data is a repository of publicly available data. It can be found at http://mocap.cs.cmu.edu/. It was generated by placing sensors on a number of

subjects and recording the position of the sensors as the subjects walked forward by a fixed amount and then walked back. There were 50 spatial dimensions of data as recorded by the sensors, which were recorded at regular times. Since the observation times were not provided, we separated each observation in time by 0.1 units and began each trajectory at time zero. The CMU Walking data which we used for our experiments consists of 75 trajectories with an average of 106.7 observations per trajectory. We could not use the full amount of data provided because some subjects did not walk in the same manner as other subjects.

It should be noted that it may not be appropriate to consider the CMU walking data as being generated by a single dynamical system. Since there were multiple subjects in the dataset, it is reasonable to conclude that there are multiple dynamical systems responsible for the motion of these different subjects. However, we fit our model to this dataset assuming that it was generated by a single dynamical system.

**Plasma**   This dataset includes imaging and plasma biomarkers from the WRAP study, see Johnson et al. (2018). The imaging biomarker is a distribution volume ratio obtained from Pittsburgh Compound-B positron emission tomography. The plasma biomarkers include $A\beta_{40}/A\beta_{42}$ that reflects specific amyloid beta ($A\beta$) proteoforms; ptau217, which has a high correlation with amyloid PET positivity, GFAP, which measures the levels of the astrocytic intermediate filament glial fibrillary acidic protein, and NFL, a recognized biomarker of subcortical large-caliber axonal degeneration, for a total of 6 biomarkers. There are a total $n = 425$ trajectories, one per subject, and, on average, 2.95 time points per trajectory. The data was split $(70\%, 10\%, 20\%)$ corresponding resp. to training, validation, and testing.

**Imaging**   This dataset includes regional imaging biomarkers from the WRAP study, see Johnson et al. (2018). The imaging biomarker is an atlas-based distribution volume ratio obtained from Pittsburgh Compound-B positron emission tomography. There are a total of 117 biomarkers. There are a total $n = 231$ trajectories, one per subject, and, on average, 2.26 time points per trajectory. The data was split $(70\%, 10\%, 20\%)$ corresponding resp. to training, validation, and testing.

## C   Comparators:

The methods we benchmark against are presented in the table below with references included. The Null

Table 3: Methods Considered

| Method | Hyperparameters |
|---|---|
| Null | N/A |
| **Sparse Identification of Nonlinear Dynamics** | |
| SINDy Brunton et al. (2016) | polynomial degree, sparsity threshold |
| **Reduced Order Models** | |
| eDMD-RFF DeGennaro & Urban (2019) | number of features, lengthscale |
| eDMD-Poly Williams et al. (2015) | polynomial degree |
| **Deep Learning** | |
| ResNet He et al. (2016) Lu et al. (2021) | network depth/breadth |
| eDMD-Deep Yeung et al. (2019) | latent space dimension, autoencoder depth/breadth |
| Latent Ode Rubanova et al. (2019) | latent space dimension, autoencoder and decoder depth/breadth |

model is used as a baseline to determine that something was learned of the dataset. Our null model has no parameters and is the trivial dynamical system for which the slope-field is zero everywhere. A model that fails to outperform the null model likely did not learn any useful information about the dynamical system.

## D   Computational Complexity

We detail below an analysis of the complexity in terms of $d$, the dimension of the state space, and $n$, the total number of observations.

### D.1 Implicit kernel

In our experiments, an implicit Gaussian kernel was used. The Gram matrix for the kernel is $n \times m$ where $n$ is the total number of samples in the dataset $X$ and $m$ is the total number of samples in the dataset $Y$. If $X \in \mathbb{R}^{d \times n}$, and $Y \in \mathbb{R}^{d \times m}$, we may compute any radial basis function kernel defined by:

$$G_{i,j} = f(\|x_i - y_j\|^2) \tag{24}$$

by observing:

$$D = \text{outer}(\text{sum}(X \circledast X), 1_m) - 2X^T Y + \text{outer}(1_n, \text{sum}(Y \circledast Y)) \tag{25}$$

Where $D_{i,j} = \|X_i - Y_j\|^2$ is the matrix of square distances, $\circledast$ is the Hadamard product, $sum$ is column sum, $outer$ is an outer product, and $1_n$ is the ones vector in $n$ dimensions. $G \in \mathbb{R}^{n \times n}$ is then calculated by applying $f$ component by component to $D$ for the training set $X$ with $X$. The computational bottleneck of this Gram matrix computation is the $n \times d$ by $d \times m$ matrix matrix multiplication, with a worst case run time that is $O(dnm) = O(dn^2)$ when $Y = X$ (with a naive implementation of matrix matrix multiplication).

We integrate over intervals of a single pair of samples, and these integrals are estimated using the trapezoid rule quadrature. Therefore, for instance, if a single trajectory is given, we would get our estimate of all our integrals by simply taking the kernel

$$k = \frac{h^2}{4} \begin{bmatrix} 1 & 1 \\ 1 & 1 \end{bmatrix} \tag{26}$$

and convolving it with the Gram matrix $G$. If multiple trajectories are given, we apply the same convolution to $G$ and ignore terms involving sums that mix samples from different trajectories. In Numpy indexing notation, we may apply the above convolution with the expression:

$$M = \frac{h^2}{4} \left( G[1:,1:] + G[:-1,1:] + G[1:,:-1] + G[:-1,:-1] \right) \tag{27}$$

This adds an additional $O(n^2)$ computational complexity (for fixed $G \in \mathbb{R}^{n \times n}$).

Finally, the linear system we solve is:

$$(M + \lambda I) A = Y \tag{28}$$

Where $M \in \mathbb{R}^{n \times n}$ and $A, Y \in \mathbb{R}^{n \times d}$, which adds the dominating computational complexity term of $O(dn^3)$.

### D.2 Explicit Kernel

Assuming the vector valued RKHS has a matrix valued kernel of the form

$$K(x, y) = \psi(x)\psi^T(y) \tag{29}$$

where

$$\psi : \mathbb{R}^d \to \mathbb{R}^{d \times q} \tag{30}$$

we may recall:

$$f(x) = \sum_i \int_{a_i}^{b_i} \psi(x)\psi(x_i(\tau))^T d\tau \alpha_i = \psi(x) \sum_i \int_{a_i}^{b_i} \psi(x_i(\tau))^T d\tau \alpha_i = \psi(x)\beta \tag{31}$$

where $\beta \in \mathbb{R}^q$ is defined by $\beta = \sum_i \int_{a_i}^{b_i} \psi(x_i(\tau))^T d\tau \alpha_i$. Letting

$$\Psi \in \mathbb{R}^{nd \times q} \tag{32}$$

be defined by:

$$\Psi_i \in \mathbb{R}^{d \times q} = \int_{a_i}^{b_i} \psi(x_i(\tau)) d\tau \tag{33}$$

we may observe that

$$L^* = \Psi\Psi^T \tag{34}$$

and

$$\alpha^T L^* \alpha = \beta^T \beta \tag{35}$$

Thus, we reduce the minimization problem 59 to

$$\min_{\beta} J(\beta) = \frac{1}{n} \sum_{i=1}^{n} \|[\Psi\beta]_i - x_i(b_i) + x_i(a_i)\|_{\mathbb{R}^d}^2 + \lambda\beta^T\beta \tag{36}$$

This simplifies to

$$\left(\Psi^T\Psi + \lambda I\right)\beta = \Psi^T y \tag{37}$$

where $y \in \mathbb{R}^{nd}$ with $y_i \in \mathbb{R}^d$ and $y_i = x_i(b_i) - x_i(a_i)$. Thus, in the explicit kernel case, we require evaluating $\Psi$ which is $O(ndq)$, a $q \times nd$ by $nd \times q$ matrix matrix multiplication $O(q^2(nd))$ and a $q \times q$ linear system solve which is $O(q^3)$. Notice, in the explicit kernel case, no assumptions on the overall structure of the kernel matrix $K(x, y)$ are needed. For instance, the matrices no longer need to be diagonal or proportional to the identity matrix. However, in practice, for best results, $q$ will typically be dependent on the dimension $d$ and the sample size.

## E  Vector Valued Reproducing Kernel Hilbert Spaces

Similarly to the scalar case, there are three ways to characterize an RKHS.

Firstly, we can construct an RKHS with linear functions of the kernel:

*Definition* 3. Let

$$H_0 = \{f; f(x) = \sum_{i=1}^{n} K(x, x_i)w_i, x_i \in \mathcal{X}, w_i \in \mathbb{R}^d, i = 1 : n\} \tag{38}$$

then, consider the encoding in $H_0$ of the functions $f$ and $g$,

$$f \longleftrightarrow \left\{ \begin{array}{l} x_1, \ldots, x_n \\ w_1, \ldots, w_n \end{array} \right. \quad \text{and } g \longleftrightarrow \left\{ \begin{array}{l} y_1, \ldots, y_m \\ v_1, \ldots, v_m \end{array} \right. \tag{39}$$

Define the inner product

$$\langle f, g \rangle = \sum_{i=1}^{n} \sum_{j=1}^{m} w_i^T K(x_i, y_j) v_j \tag{40}$$

then the closure of $H_0$ for $\langle ., . \rangle$ is the RKHS of K.

The second construction starts from a Hilbert space and uses the Riesz representation theorem.

*Definition* 4. Let $H$ be a Hilbert space of functions $\mathbb{R}^d \to \mathbb{R}^d$ such that for any $x \in \mathbb{R}^d$, the evaluation functional $f \mapsto f(x)$ is continuous: there is a constant $M_x \in \mathbb{R}$, such that

$$\|f(x)\|_{\mathbb{R}^d} \le M_x \|f\|_H \tag{41}$$

then, using the Riesz representation, for any $v \in \mathbb{R}^d$ there exists a unique $K_{x,v} \in H$ such that

$$v^T f(x) = \langle f, K_{x,v} \rangle_H \tag{42}$$

This equation is the *reproducing property*.

Define the matrix-valued kernel

$$K_{ij}(x, x') = \langle K_{x,e_i}, K_{x',e_j} \rangle, i, j = 1 \ldots d \tag{43}$$

where $e_1, \ldots, e_d$ is the natural basis of $\mathbb{R}^d$. Then $K$ is a kernel and $H$ is the RKHS of $K$.

The third formulation allows for checking that a Hilbert space is an RKHS.

*Definition* 5. Let $H$ be a Hilbert space of functions $\mathbb{R}^d \to \mathbb{R}^d$. Let $K$ (d,d) be a kernel. H is the RKHS of K when

1. For any $x' \in \mathcal{X}$, $v \in \mathbb{R}^d$, $x \mapsto K(x, x')v \in H$

2. The reproducing property holds: for any $f \in H$, $x \in \mathcal{X}$, $v \in \mathbb{R}^d$, equation 42

### E.1 Verifying the continuity for the occupation kernel in multiple dimensions

$(e_1, \ldots, e_d)^T$ is the standard basis of $\mathbb{R}^d$.

$$|e_j^T L_x(f)| = \left| \int_0^T e_j^T f(x(t)) dt \right| \tag{44}$$

$$\leq \int_0^T \left| e_j^T f(x(t)) \right| dt \tag{45}$$

$$= \int_0^T |\langle f, K(., x(t)) e_j \rangle| \, dt \tag{46}$$

$$\leq \|f\| \int_0^T \sqrt{K_{jj}(x(t), x(t))} dt \tag{47}$$

where we have used the Cauchy-Schwartz inequality. Since $t \mapsto x(t)$ is continuous, if moreover $x \mapsto K_{jj}(x, x)$ is continuous for each $j = 1 \ldots d$, then for each $x$ and each $1 \leq j \leq d$, $e_j^T L_x$ is bounded which implies that $L_x$ is bounded and thus continuous.

## F  Proof of Theorem 1

Consider the linear span of the occupation kernel functions $L_{x_i}^*$, $i = 1 \ldots n$

$$\mathcal{F} = \{ f \in \mathbb{H}, f = \sum_{i=1}^n L_{x_i}^*(\alpha_i), \alpha = (\alpha_1, \ldots, \alpha_n) \in \mathbb{R}^{dn} \} \tag{48}$$

Note that $\mathcal{F}$ is linear and finite-dimensional; thus, it is a closed linear subspace of $\mathbb{H}$. We can then project any function in $\mathbb{H}$ orthogonally onto it

$$f = f_{\mathcal{F}} + f_{\mathcal{F}^\perp} \tag{49}$$

Now, for each $i = 1 \ldots n$, and $v \in \mathbb{R}^d$,

$$v^T \int_{a_i}^{b_i} f(x_i(t)) dt = v^T L_{x_i}(f) = \langle L_{x_i}^*(v), f \rangle = \langle L_{x_i}^*(v), f_{\mathcal{F}} + f_{\mathcal{F}^\perp} \rangle = \langle L_{x_i}^*(v), f_{\mathcal{F}} \rangle \tag{50}$$

where the previous to last equality comes from the fact that $f_{\mathcal{F}^\perp}$ is perpendicular to $L_{x_i}^*(v)$. Thus,

$$\int_{a_i}^{b_i} f(x_i(t)) dt = L_{x_i}(f_{\mathcal{F}}) \tag{51}$$

Next, using the Pythagorean equality

$$\|f\|^2 = \|f_{\mathcal{F}}\|^2 + \|f_{\mathcal{F}^\perp}\|^2 \geq \|f_{\mathcal{F}}\|^2 \tag{52}$$

Thus, for any $f \in \mathbb{H}$, $J(f_{\mathcal{F}}) \leq J(f)$ which proves that the minimum of $J$ actually belongs to $\mathcal{F}$. Moreover, note that

$$||f_{\mathcal{F}}||^2 = \left\langle \sum_{i=1}^n L_{x_i}^*(\alpha_i), \sum_{i=1}^n L_{x_i}^*(\alpha_i) \right\rangle = \sum_{i,j=1}^n \left\langle L_{x_i}^*(\alpha_i), L_{x_j}^*(\alpha_j) \right\rangle$$

$$= \sum_{i,j=1}^n \alpha_i^T \left[ \int_{a_i}^{b_i} \int_{a_j}^{b_j} K(x_i(s), x_j(t)) ds dt \right] \alpha_j = \sum_{i,j=1}^n \alpha_i^T L_{ij}^* \alpha_j = \alpha^T L^* \alpha \quad (53)$$

Also,

$$L_{x_i}(f_{\mathcal{F}}) = L_{x_i} \left( \sum_{j=1}^n L_{x_j}^*(\alpha_j) \right) \tag{54}$$

$$= \int_{a_i}^{b_i} \sum_{j=1}^n L_{x_j}^*(\alpha_j)(x_i(t)) dt \tag{55}$$

$$= \sum_{j=1}^n \left[ \int_{a_i}^{b_i} \int_{a_j}^{b_j} K(x_i(t), x_j(s)) ds dt \right] \alpha_j \tag{56}$$

$$= \sum_{j=1}^n L_{ij}^* \alpha_j \tag{57}$$

$$= [L^* \alpha]_i \tag{58}$$

The minimization problem in $f$ is thus equivalent to minimizing in $\alpha$

$$J(\alpha) = \frac{1}{n} \sum_{i=1}^n \|[L^*\alpha]_i - x_i(b_i) + x_i(a_i)\|_{\mathbb{R}^d}^2 + \lambda \alpha^T L^* \alpha \tag{59}$$

or equivalently,

$$J(\alpha) = \frac{1}{n} \|L^*\alpha - x(b) + x(a)\|_{\mathbb{R}^{nd}}^2 + \lambda \alpha^T L^* \alpha \tag{60}$$

since $\lambda > 0$ and $L^*$ is psd, this is solved by

$$(L^* + \lambda n I) \alpha = x(b) - x(a) \tag{61}$$

## G  Vector field learned from the buoy data

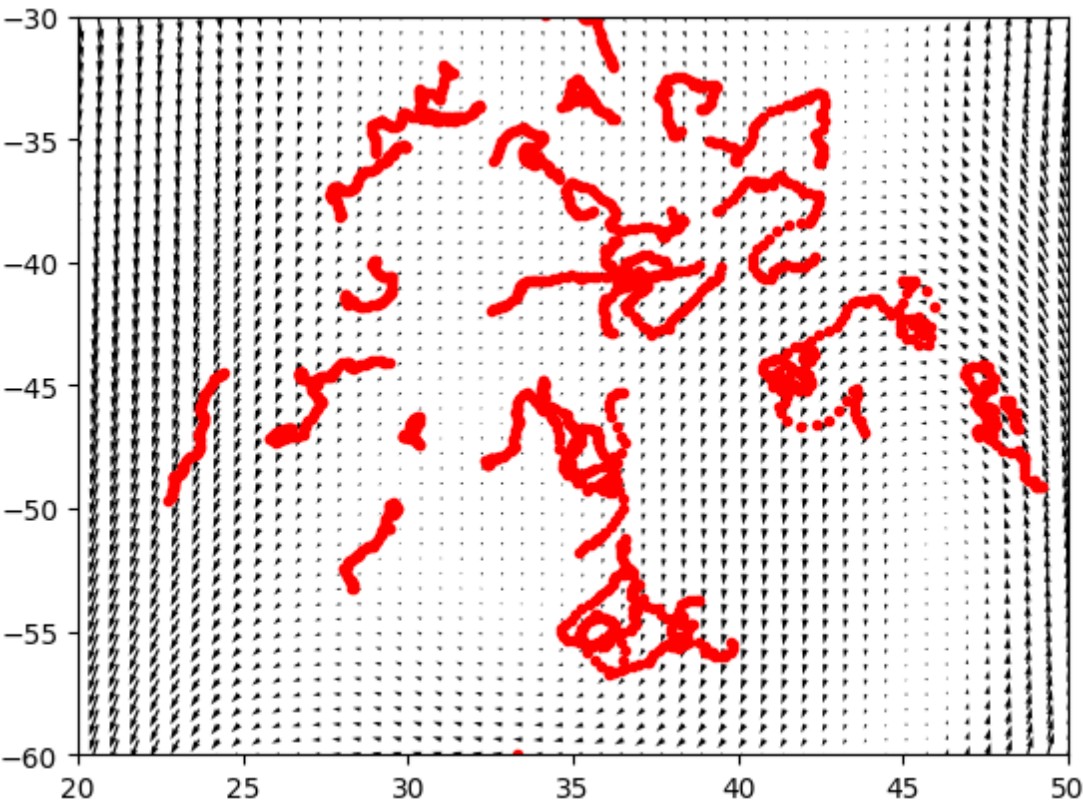

Figure 5: A detail of the vector field learned using the divergence-free kernel from the buoy data together with the training set (red).

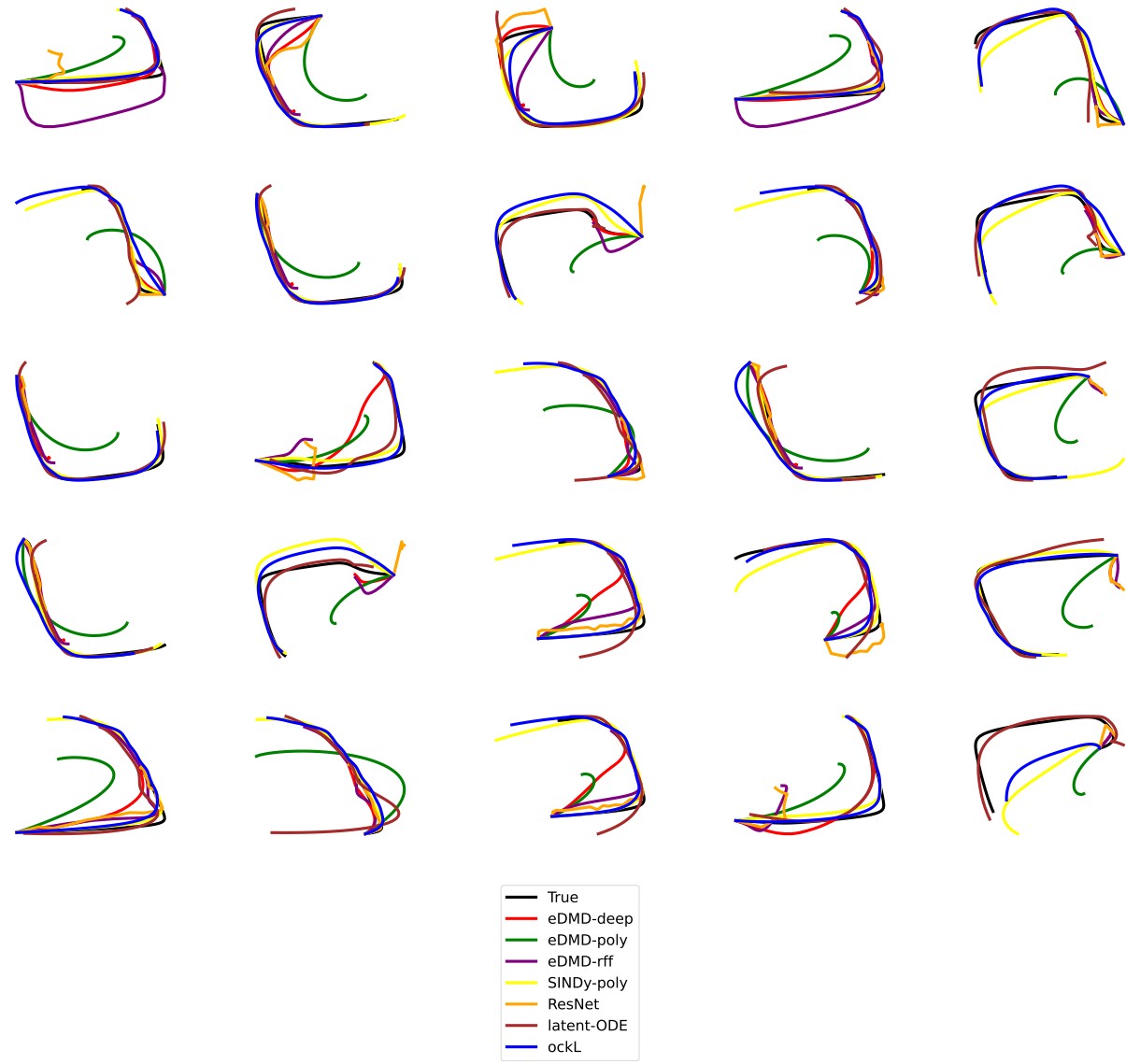

Figure 6: These are 25 randomly sampled trajectories in the test set for FHN. Black is the true trajectory, red is eDMD-deep, green is eDMD-poly, purple is eDMD-rff, yellow is SINDy-poly, orange is ResNet, brown is latent-ODE, and blue is ockL.

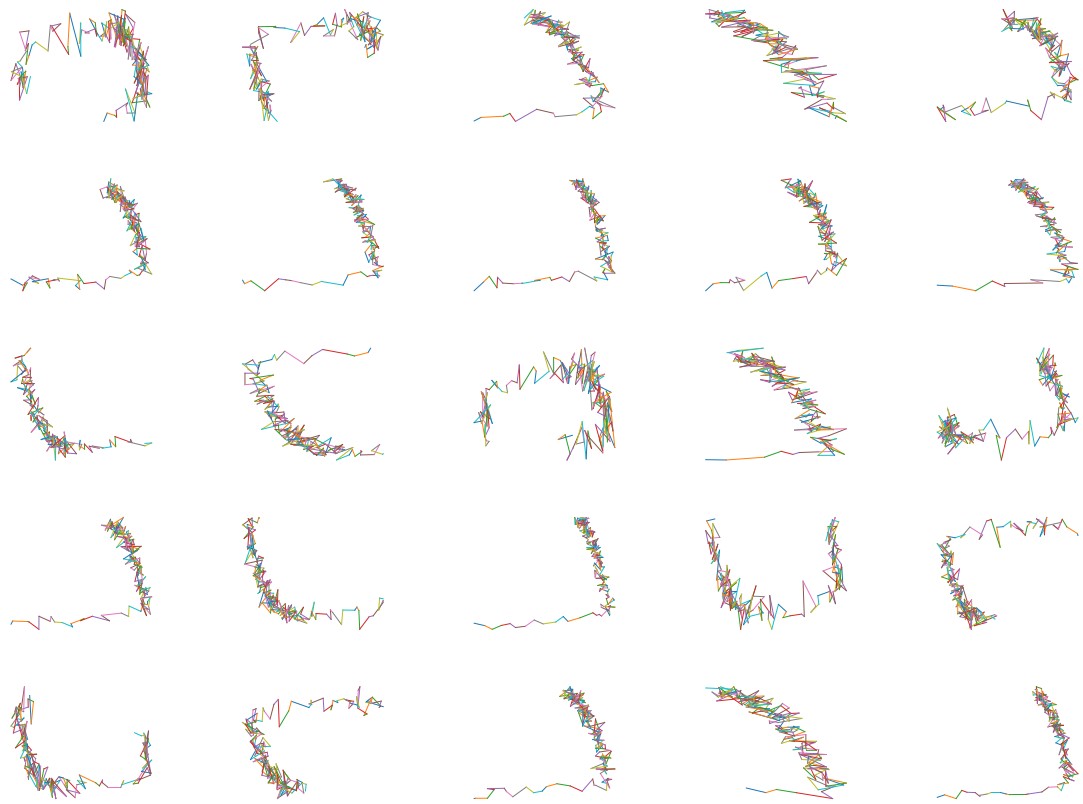

Figure 7: 25 randomly sampled trajectories in the NFHN (Noisy FHN) training set.

