# OpenReview forum: "Learning Nonparametric Differential Equations via Multivariate Occupation Kernel Functions"
_TMLR — Rejected by TMLR_

### Review · Reviewer_FAJB · 2024-09-02

**Summary Of Contributions:**

Overall, this work focused on learning a nonparametric system of ordinary differential equations from trajectories and proposed a novel method based on the occupation kernel method with Reproducing Kernel Hilbert Spaces, which has only linear complexity in the dimension d. In addition, experimental results support the efficiency of the proposed method.

**Audience:**

Yes

**Broader Impact Concerns:**

N/A.

**Claims And Evidence:**

Yes

**Requested Changes:**

See Weakness.

**Strengths And Weaknesses:**

Strengths:

1. The paper is well-written and easy to follow.

2. Experimental results support the efficiency of the proposed method.

Weaknesses:

1. Although the authors claim linear complexity in the dimension d, there is an additional dependency on n and q (as discussed in Section 3.4). The linear dependency on d becomes less meaningful if the factors n and q are larger than d, which is often the case in Reproducing Kernel Hilbert Spaces. It is necessary to provide the complexity of other frameworks to make the comparison clearer.

2. While the algorithm claims low computational complexity, this is not supported by the experimental results. It would be beneficial to provide the running time for different methods in a table for further comparison.

---

> ### Author Response · Authors · 2024-10-31
> **Review Response**
>
> We carefully considered the reviews and prepared a thorough response we hope will answer your questions and clarify any changes we will be making to the submission in response to your concerns. The responses to all reviews may be found as a pdf file at:
>
> https://drive.google.com/file/d/1HMsBVf6ff0bu83Kz-w30d0lxsqizJGBb/view?usp=sharing

---

### Review · Reviewer_MBZQ · 2024-09-24

**Summary Of Contributions:**

This submission aims to solve the trajectory recovery problem based on ODE through formulating the problem as solving a linear system. The coefficient matrix of the linear system is given by _occupation kernel function_ of observed curve points.

**Audience:**

Yes

**Claims And Evidence:**

No

**Requested Changes:**

Questions:

- In equation (6), if $L^*$ means adjoint operator, it does not make sense to me that $w$ and $v$ are not from the same space while the property of adjoint operators still applies.

- What are the relationship between $n$ and $d$? In other words, how many data points we need to make the computation to be accurate enough? What happens when $n\ll d$? What happens when $n\gg d$?

- explain what kernels are used to set up the linear system for trajectory recovery with quantitative details.

- How does one choose $\lambda$?

- explain the dataset: what are the raw data? What are corresponding input and output if parametric models are used in any baseline method?

- Does each method see the same input data? Please explain what the raw data is for each baseline method and proposed method is respectively.

- What is the kernel satisfying the energy conservation law, in section 5.2?

**Strengths And Weaknesses:**

Weaknesses:

- The writing of the manuscript can be largely improved. I appreciate authors' succinct style with which the argument cuts directly into the main topic. Meanwhile, it is equivalently important to ensure the clarity and completeness of all relevant definitions, notations, theorem statement, and algorithm input and output conditions.

---

> ### Author Response · Authors · 2024-10-31
> **Review Response**
>
> We carefully considered the reviews and prepared a thorough response we hope will answer your questions and clarify any changes we will be making to the submission in response to your concerns. The responses to all reviews may be found as a pdf file at:
>
> https://drive.google.com/file/d/1HMsBVf6ff0bu83Kz-w30d0lxsqizJGBb/view?usp=sharing

---

> ### Comment · Reviewer_MBZQ · 2024-11-05
> **More specification about $n$ and $d$ relation**
>
> I acknowledge authors kind effort to reply to my questions.
>
> Authors can incorporate information provided in this reply into main text to make the narrative much clearer. Polishing the writing can improve the readability of the manuscript for the general community.
>
> Authors state that the specific discussion about $n$ and $d$'s impact to the accuracy of this algorithm is left for future work. At least for this submission, numerical details should be very explicitly given about what $n$ and $d$ are in this work, and what the main reason was for authors to choose that combination.
>
> Meanwhile, I would like to see authors specify
> - How the integral quadrature is computed?
> - What solver is used to solve the linear system? What is the corresponding parameter setting used for the solver?

---

> > ### Author Response · Authors · 2024-11-18
> > **Response to additional specifications**
> >
> > Thank you for your comments.
> > We will add a comment about the numerical error of the integration in terms of n and d, based on our response to your questions.
> >  * How the integral quadrature is computed?
> > We use the trapezoid rule to integrate the trajectories and we will make that clear in the paper.
> > * What solver is used to solve the linear system? What is the corresponding parameter setting used for the solver?
> > We use python's numpy linear system solver "np.linalg.solve" (numpy version 1.26.4). We use the default settings.

---

### Review · Reviewer_8gNz · 2024-10-18

**Summary Of Contributions:**

This work extends the Occupation Kernel method for inference in a kind of trajectory space. The algorithm favourably in the dimension of the ambient space of the trajectory. By finding finite-dimensional data-driven representation, we can recover estimates of a forcing field influencing the noisily-observed trajectories. Using some neat kernel tricks, the field can even be forced to satisfy physically-interesting constraints (e.g. being divergence free)

**Audience:**

Yes

**Claims And Evidence:**

Yes

**Requested Changes:**

## General explanation

I mentioned a few "weaknesses" above which I submit could each be clarified in the paper. Specifically I   think it would be improved if it clarified

* generally taking more time to build intuitions in the introduction, esp 2.3 and 3.2
* the point of difference between the earlier OCK works by Rosenfeld and Russo
* why the unique regularised RKHS-norm minimizer might corresponds to the true field of interest

## Computational complexity claims

In 1.4 we learn that this paper claims to improve on the computational complexity of the Rosenfeld and Russo papers. I can't actually find the complexity claims in any of those papers; perhaps I'm missing something? But the claim should be clearer; if the scaling is better than the competition, compare them side-by-side

## Clarity point 1

>Let us now assume that instead of observing $n$ curves that are solutions of equation 9 , we observe $m+1$ snapshots _of each curve_, sometimes noisy, $z_i=\left(z_i^{(0)}, \ldots, z_i^{(m)}\right)$ at time-points $t_0, \ldots, t_m$ coming from a true trajectory $x_i$, $i=1 \ldots p$. Firstly, we reshape this data into observations coming from $n=m p$ trajectories, each made of two samples. This is done by viewing every couple of consecutive observations as two observations associated with a single trajectory. In other words, we assume that we observe (possibly with noise) the initial and final points of $n$ trajectories $x_i, i=1 \ldots n$, that we denote by $z_i=\left(z_i^{(0)}, z_i^{(1)}\right) . z_i$ can therefore be viewed as a matrix of dimensions $d$ by 2 .

I'm having a hard time parsing this. Can you confirm that I have got it right in this rephrasing?

>Let us now assume that instead of observing $n$ curves that are solutions of equation 9, for each true trajectory $x_i$, we observe $m+1$ noisy snapshots, $z_i=\left(z_i^{(0)}, \ldots, z_i^{(m)}\right)$  at time-points $t_0, \ldots, t_m$ where
>$z_i^{(j)}=x_i(t_m)+\epsilon_i^{(m)}, \epsilon \sim \mathcal{N}(0,\sigma^2\mathrm{I}_d)$ i.e.
> $\\{x\_i(t)\\}\_{t\in [t\_{j-1}, t\_j]}, i=1,\dots,p, j=1,\dots, m$.
> By renumbering the curves we treat these as  $n=m p$ separate trajectories, where each is characterised by initial and final state, $z_i=\left(z_i^{(0)}, z_i^{(1)}\right) .$  $z_i$ can therefore be viewed as $d\times 2$ matrix

## Clarity point 2

eq (6) the definition of the occupation kernel kernel function $L_x^*(v)$ does a lot of work, so I think the authors could invest effort in clarifying it. AFAICT

$$
\begin{aligned}
L_x: & \mathbb{H}&\to \mathbb{H}\\\\
L_x^*: & \mathbb{R}^d&\to \mathbb{H}
\end{aligned}
$$

I spent a long time thinking the $\cdot ^*$ was intended to suggest a complex conjugate or something on functions between the same spaces. Does it need to recycle the same letter? This notation choice is unintuitive enough that it could be changed or at least explained.


## Minor points

Sec 3.3: "Integral quadrature" -> "numerical quadrature" (or am I missing something?)

Section 3.3 "every couple of consecutive observations" -> "every pair of consecutive observations"

App C title "Comparators:" -> "Comparators", i.e. delete colon

**Strengths And Weaknesses:**

# Strengths

The paper presents an interesting method which satisfyingly solves a tricky problem in inference of spatial fields, which is a difficult, and high-value problem.
From the experiments it seems to be remarkably good at identifying function-valued estimands from sparse data, which is an important and impressive problem.

# Weaknesses

These may be mostly weaknesses in my own understanding; let us find out.


## brusqueness

In TMLR we have a little more space than in conferences to expand our explanations; we have room for more intuition building. This paper does not make use of those, so the introduction is very condensed. Especially for less well-known methods (I had not heard of the OCK before this) it would be good to use that space to introduce the concepts to the reader

## Contribution specifics

It is hard to extract from the paper what exactly the unique contribution of this paper is compared to the antecedents. The OCK method, whcih I learned about from this paper, looks useful, and I am very open to it, but also, I suspect that I am like many readers in that I don't know much about it. In particular, I don't know exactly what is standard, and what is novel in this paper, and this paper does not make it easy for me to discover that; instead I have the Rosenthal et al and Russo et al papers side-by-side on my screen to compare item-by-item (not that this excuses my slow review, for which I must apologise)

## Uniqueness of solution

The paper plays the standard trick (Theorem 3.1) of finding unique regularised $L_2$ minimisers in the RKHS, which seems sound.  But the reader (well, _this_ reader) is left with a problem of intuition-building: how do we connect the unique RKHS solution with the notorious non-uniquness of inverse problems generally? In a naive attempt to solve similar problems (e.g. directly inferring trajectories by path derivatives) I expect to find many paths compatible with the solution. I have spent some time trying to understand this phenomenon, but got nowhere. Can you build intuitions as to how the OCK trick escapes this problem?

## Section 5 and existing literature on constrained GPs

Section 5 introduces an interesting elaboration of the basic algorithm, constraining the solutions to an admissible function space.
Notably missing is the work on RKHSs and in particular GPs over interesting function spaces constrained by linear operators. The authors' work in this paper could well be a novel approach (since the OCK is not quite the classic solution), but also, physically-constrained kernels is one of those areas that is perpetually "rediscovered" and thus always worth checking the literature for precedent. I suspect that some of the below references also cite Fuselier (2008) as invoked here, in which case my point may be moot.

A partial bibliography of such  methods:

```
@article{AlbertGaussian2019,
  title = {Gaussian {{Processes}} for {{Data Fulfilling Linear Differential Equations}}},
  author = {Albert, Christopher G.},
  year = {2019},
  month = nov,
  journal = {Proceedings},
  volume = {33},
  number = {1},
  pages = {5},
  doi = {10.3390/proceedings2019033005}
}

@inproceedings{BesginowConstraining2024,
  title = {Constraining {{Gaussian}} Processes to Systems of Linear Ordinary Differential Equations},
  booktitle = {Proceedings of the 36th {{International Conference}} on {{Neural Information Processing Systems}}},
  author = {Besginow, Andreas and {Lange-Hegermann}, Markus},
  year = {2024},
  month = apr,
  series = {{{NIPS}} '22},
  volume = {35},
  pages = {29386--29399},
  publisher = {Curran Associates Inc.},
  address = {Red Hook, NY, USA},
  isbn = {978-1-71387-108-8}
}

@inproceedings{BolinEfficient2021,
  title = {Efficient Methods for {{Gaussian Markov}} Random Fields under Sparse Linear Constraints},
  booktitle = {Advances in {{Neural Information Processing Systems}}},
  author = {Bolin, David and Wallin, Jonas},
  year = {2021},
  volume = {34},
  pages = {9882--9894},
  publisher = {Curran Associates, Inc.}
}

@article{GulianGaussian2022,
  title = {Gaussian Process Regression Constrained by Boundary Value Problems},
  author = {Gulian, M. and Frankel, A. and Swiler, L.},
  year = {2022},
  month = jan,
  journal = {Computer Methods in Applied Mechanics and Engineering},
  volume = {388},
  pages = {114117},
  doi = {10.1016/j.cma.2021.114117}
}

@inproceedings{HarkonenGaussian2023,
  title = {Gaussian {{Process Priors}} for {{Systems}} of {{Linear Partial Differential Equations}} with {{Constant Coefficients}}},
  booktitle = {Proceedings of the 40th {{International Conference}} on {{Machine Learning}}},
  author = {Harkonen, Marc and {Lange-Hegermann}, Markus and Raita, Bogdan},
  year = {2023},
  month = jul,
  pages = {12587--12615},
  publisher = {PMLR}
}

@article{HendersonCharacterization2023,
  title = {Characterization of the Second Order Random Fields Subject to Linear Distributional {{PDE}} Constraints},
  author = {Henderson, Iain and Noble, Pascal and Roustant, Olivier},
  year = {2023},
  month = nov,
  journal = {Bernoulli},
  volume = {29},
  number = {4},
  pages = {3396--3422},
  publisher = {{Bernoulli Society for Mathematical Statistics and Probability}},
  doi = {10.3150/23-BEJ1588}
}

@phdthesis{HendersonPDE2023,
  title = {{{PDE}} Constrained Kernel Regression Methods},
  author = {Henderson, Iain},
  year = {2023},
  month = sep,
  school = {INSA de Toulouse}
}

@inproceedings{HutchinsonVectorvalued2021a,
  title = {Vector-Valued {{Gaussian Processes}} on {{Riemannian Manifolds}} via {{Gauge Independent Projected Kernels}}},
  booktitle = {Advances in {{Neural Information Processing Systems}}},
  author = {Hutchinson, Michael and Terenin, Alexander and Borovitskiy, Viacheslav and Takao, So and Teh, Yee and Deisenroth, Marc},
  year = {2021},
  volume = {34},
  pages = {17160--17169},
  publisher = {Curran Associates, Inc.}
}

@inproceedings{LangeHegermannAlgorithmic2018,
  title = {Algorithmic Linearly Constrained {{Gaussian}} Processes},
  booktitle = {Proceedings of the 32nd {{International Conference}} on {{Neural Information Processing Systems}}},
  author = {{Lange-Hegermann}, Markus},
  year = {2018},
  month = dec,
  series = {{{NIPS}}'18},
  pages = {2141--2152},
  publisher = {Curran Associates Inc.},
  address = {Red Hook, NY, USA}
}

@inproceedings{LangeHegermannLinearly2021,
  title = {Linearly Constrained Gaussian Processes with Boundary Conditions},
  booktitle = {Proceedings of the 24th International Conference on Artificial Intelligence and Statistics},
  author = {{Lange-Hegermann}, Markus},
  editor = {Banerjee, Arindam and Fukumizu, Kenji},
  year = {2021},
  month = apr,
  series = {Proceedings of Machine Learning Research},
  volume = {130},
  pages = {1090--1098},
  publisher = {PMLR}
}
```

---

> ### Author Response · Authors · 2024-10-31
> **Review Response**
>
> We carefully considered the reviews and prepared a thorough response we hope will answer your questions and clarify any changes we will be making to the submission in response to your concerns. The responses to all reviews may be found as a pdf file at:
>
> https://drive.google.com/file/d/1HMsBVf6ff0bu83Kz-w30d0lxsqizJGBb/view?usp=sharing

---

> > ### Comment · Reviewer_8gNz · 2024-11-04
> > **Got it.**
> >
> > OK, that was an excellent response, and generally quite helpful
> >
> > One small point:
> >
> > I thank the authors for the excellent answer to my question
> >
> > >>why the unique regularised RKHS-norm minimizer might correspond to the true field of interest
> >
> >
> > specifically,
> >
> > >To build some intuition about how we arrive at the uniqueness of the vector field minimizing the cost function
> > in equation (10), first, recall that the first term in the cost function relates to fitting the observed trajectories.
> > >For simplicity, assume that you have multiple solutions (functions in the RKHS) that fit the data perfectly, i.e.,
> > setting that term to zero.
> > >Now, consider Occam’s razor principle, which states that, among competing hypotheses that explain the data
> > equally well, the simplest one should be chosen. In this context, simplicity corresponds to the smoothness of the
> > solution, which is measured by the RKHS norm. This is where the second term of the cost function comes into
> > play. According to Occam’s razor, we should select the solution with the smallest RKHS norm among those that
> > fit the data perfectly, as it represents the smoothest solution.
> > >One might argue that there could be multiple solutions with the same minimal norm that fit the data perfectly.
> > >However, this is impossible because if that were the case, any convex combination of those solutions would have
> > a smaller norm - since the function we optimize is strictly convex - and would still fit the data perfectly. This
> > would contradict the assumption that the original solutions had minimal norm, thus proving that such a scenario
> > cannot occur. Therefore, the minimizer is unique.
> >
> > I perhaps should have been more clear; I did not necessarily wish to have the authors explain it to _me_ in detail; rather I was suggesting that motivating this in the text would benefit the exposition. I was after an extra sentence or two in the text body.
> >
> > This is, to be clear, not a blocker for me

---

> > > ### Author Response · Authors · 2024-11-18
> > > **Agreed**
> > >
> > > Agreed we will add a summary of the provided answer to the paper to improve the exposition. Thank you for your helpful review.

---

### Decision · Action_Editor_Mnad · 2024-11-25

**Recommendation:** Reject

**Comment:**

While the paper addresses a relevant problem and has potential, the concerns raised by **Reviewer MBZQ** are critical and must be addressed before the work can be published. The lack of clarity, incomplete exposition, and missing methodological details hinder the understanding and reproducibility of the research. These issues are significant enough to warrant a rejection at this time.

**Recommendations:**

- Revise the manuscript to enhance the clarity of the exposition. Ensure that all definitions, notations, and theorem statements are clearly presented. Incorporate the explanations and clarifications provided in the authors' responses into the main text.

- Provide detailed and consistent information about hyperparameter choices, including the relationship between key parameters and their impact on the algorithm's accuracy.

- Include essential details about the computational methods used, such as how the integral quadrature is computed and specifics about the solver for the linear system.

- Strengthen the experimental section by providing computational complexity comparisons (a major claim) with other frameworks to substantiate claims about efficiency.

As there is no major revision option in the system at the current stage, we can only select the reject option but highly encourage the authors to thoroughly revise the manuscript and resubmit the work in the future.

**Audience:**

This paper aims to learn differential equations from data, which is relevant to the TMLR audience.  However, due to problems with demonstration and experimental validation, current submissions do not fully satisfy the audience's interest.

**Claims And Evidence:**

While the submission presents interesting ideas and has potential After reviews, reviewers still have some concerns related to the completeness of the explanation, inconsistencies in hyperparameter settings, and lack of essential methodological details. These shortcomings impede the understanding and reproducibility of the work.

**Resubmission Of Major Revision:**

The authors may consider submitting a major revision at a later time.